# Polarization of Microglia and Its Therapeutic Potential in Sepsis

**DOI:** 10.3390/ijms23094925

**Published:** 2022-04-28

**Authors:** Léo Victor G. Castro, Cassiano F. Gonçalves-de-Albuquerque, Adriana R. Silva

**Affiliations:** 1Laboratório de Imunofarmacologia, Instituto Oswaldo Cruz, Fundação Oswaldo Cruz (FIOCRUZ), Rio de Janeiro 21040-900, Brazil; leovictorgrimaldidecastro@gmail.com; 2Programa de Pós-Graduação em Biologia Celular e Molecular, Instituto Oswaldo Cruz, Fundação Oswaldo Cruz (FIOCRUZ), Rio de Janeiro 21040-900, Brazil; 3Laboratório de Imunofarmacologia, Universidade Federal do Estado do Rio de Janeiro (UNIRIO), Rio de Janeiro 20211-010, Brazil; 4Programa de Pós-Graduação em Biologia Molecular e Celular, Universidade Federal do Estado do Rio de Janeiro (UNIRIO), Rio de Janeiro 20211-010, Brazil

**Keywords:** neuroinflammation, microglia, PPARγ, molecular targets

## Abstract

Sepsis is a life-threatening organ dysfunction caused by a dysregulated host response to infection, leaving the inflammation process without a proper resolution, leading to tissue damage and possibly sequelae. The central nervous system (CNS) is one of the first regions affected by the peripheral inflammation caused by sepsis, exposing the neurons to an environment of oxidative stress, triggering neuronal dysfunction and apoptosis. Sepsis-associated encephalopathy (SAE) is the most frequent sepsis-associated organ dysfunction, with symptoms such as deliriums, seizures, and coma, linked to increased mortality, morbidity, and cognitive disability. However, the current therapy does not avoid those patients’ symptoms, evidencing the search for a more optimal approach. Herein we focus on microglia as a prominent therapeutic target due to its multiple functions maintaining CNS homeostasis and its polarizing capabilities, stimulating and resolving neuroinflammation depending on the stimuli. Microglia polarization is a target of multiple studies involving nerve cell preservation in diseases caused or aggravated by neuroinflammation, but in sepsis, its therapeutic potential is overlooked. We highlight the peroxisome proliferator-activated receptor gamma (PPARγ) neuroprotective properties, its role in microglia polarization and inflammation resolution, and the interaction with nuclear factor-κB (NF-κB) and mitogen-activated kinases (MAPK), making PPARγ a molecular target for sepsis-related studies to come.

## 1. Introduction

Sepsis is a life-threatening organ dysfunction caused by a dysregulated host response to infection [1]. Even though the inflammation process is one of the main strategies the innate immune system uses to neutralize a pathogen, effectively controlling local infections, the same can be harmful without a proper response during a disseminated infection [2]. The host response starts with the recognition of danger-associated molecular patterns (DAMPs) and pathogens-associated molecular patterns (PAMPs), activating the innate immune cells, mainly monocytes, macrophages, neutrophils, and natural killer cells [3]. Pathogen recognition will lead to the activation of intracellular transduction signal pathways, elevating the release of pro-inflammatory cytokines, which will subsequently activate the coagulation and complement pathways, affecting leucocyte activation [4]. This process, when deregulated, can lead to extensive tissue damage and physical and psychological sequelae, along with long-term cognitive impairment and functional disability [5]. The estimated incidence is 48.9 million cases, with 11 million sepsis-related deaths in 2017, representing 19.7% of global deaths. These numbers are considerably higher in areas with low and middle socio-demographic indexes [6]. The burden of sepsis is so substantial that in 2017, the World Health Organization and World Health Assembly recognized it as a global health priority, adopting an action plan for all United Nations Members, focusing on improving the prevention, diagnosis, and treatment of sepsis [7]. 

The central nervous system (CNS) is one of the first regions exposed to an inflammatory episode due to the progress of peripheral inflammation caused by sepsis [8]. The inflammatory signals reach the brain through the humoral and the neural pathways, which depend on the breach of the blood–brain barrier and the activation of the vagus nerve fibers, respectively [9], as in neurodegenerative diseases. This process allows peripheral inflammatory mediators and immune cells into the brain, also activating the resident immune cells of the CNS, exposing the neurons to an environment of oxidative stress, leading to neuronal dysfunction and apoptosis [10]. Sepsis-associated encephalopathy (SAE) occurs in up to 70% of all septic patients, constituting the most frequent organ dysfunction associated with sepsis and the leading cause of brain dysfunction in intensive care units (ICU) [11]. The most common symptoms associated with SAE are deliriums, seizures, and coma, related to increased mortality, morbidity, and cognitive disability [12,13]. In addition, around 25 to 50% of the survivors will experience risks of developing considerable neurocognitive impairment, such as problems with memory, concentrating, and decision-making [14]. The cerebral damage caused by sepsis may also affect one’s mental health, where the surviving patients are commonly diagnosed with post-traumatic stress disorder (PTSD), anxiety, and depression [15]. 

To decrease the risk of sepsis long-term sequelae, the current guidelines for sepsis management focus on early recognition and treatment. These recommendations revolve around initial fluid resuscitation of patients with sepsis-induced hypoperfusion and high lactate levels, enhanced screening and performance improvement, routine microbiological cultures for proper diagnosis, and a broad-spectrum therapy with one or more antimicrobials to cover all pathogens’ possibilities [1]. However, surviving patients still experience cognitive impairment and mental health issues, denoting brain impairment and evidencing the need for a more optimal approach [16]. The study of cellular and molecular mechanisms involved in physiopathological processes causing SAE will guide the new therapy advances. Herein, we focus on the protective role of microglia polarization, discussing the effects of synthetic compounds and natural PPARγ agonists and PPARγ interactions with transcription factors and intracellular signaling proteins, highlighting PPARγ as potential target for non-infectious and infectious disease adjuvant therapy. The pleiotropic effects of PPARγ agonists make them great candidates for therapy associated with antibiotics in sepsis.

## 2. Microglia and Their Role in the CNS

Microglia cells originate from yolk sac-derived macrophage progenitors that migrate to the brain at the early stages of embryonic development before the blood–brain barrier is formed [17]. They can self-renew and exhibit one the most extended life-spans of any myeloid cells in the body microglia represent the main defense line in the CNS, playing a pivotal role in maintaining CNS homeostasis and neurological function [18,19,20]. They can control neurogenesis by phagocytosis and elimination of dying neurons. In addition, microglia secrets soluble factors, such as the nerve growth factor (NGF) and the tumor necrosis factor (TNF), regulating neuronal apoptosis [21]. Microglia is involved in the elimination and synaptic connectivity refinement by engulfing pre and postsynaptic structures [22]. Microglia secretes neurotrophic factors, brain-derived neurotrophic factor (BDNF) being the most important, with a central role in synaptic plasticity and affecting the structure and function of adjacent synapses [23].

The systemic immune activation observed during sepsis alters microglia behavior, with a pro-inflammatory profile [24]. The role of the M1 subtype, pro-inflammatory microglia, mainly revolves around the neutralization of an invading pathogen by the production of pro-inflammatory cytokines TNF-α, interleukin-1 beta (IL-1β), interleukin-6 (IL-6), and interleukin-2 (IL-12), nitric oxide (NO), reactive oxygen species (ROS), and superoxide [25]. Dysregulated M1 microglia response plays a significant part in SAE and sepsis-associated chronic pain [26]. Once the pro-inflammatory response is established, microglia start shifting into a more neuroprotective and anti-inflammatory phenotype. M2 microglia plays a crucial role in resolving inflammation, toxicity clearance, and preserving brain tissue [27]. These cells can be further divided into subgroups, each with its unique function in the CNS. M2a is involved in repair (mainly by removal of cell debris) and regeneration, M2b in immune regulation, and M2c in neuroprotection and the production of anti-inflammatory cytokines, such as interlukin-10 (IL-10) and interleukin-4 (IL-4) [28]. This switch between both phenotypes allows microglia to start and suppress the neuroinflammation process, protecting the brain tissue and maintaining CNS homeostasis without harming its integrity [29]. Any disturbance, such as that observed during sepsis, can lead to neuronal death and function impairment. 

The neurons damaging and killing by microglia are often related to neurodegenerative diseases, with the M1 phenotype mostly viewed as detrimental. Microglia activity is heavily involved with most of CNS diseases, consequently having a substantial impact on their outcome. Microglia express a variety of cell surface receptors, such as Toll-like receptors (TLRs), phagocytic receptors (CR3, CR4), and scavengers’ receptors (CD36, CD91), all upregulated when in response to injury or infections [30]. These cells respond to DAMPs from damaged, stressed, or dying cells and PAMPs, activating microglia receptors, leading to M1-related response [31]. Pro-inflammatory cytokines secreted by M1 microglia are associated with brain damage [32]. Upregulation of TNF-α secretion is linked to endothelial necroptosis and increased BBB permeability, perpetuating neuroinflammation [33]. Increased levels of TNF-α can also intensify glutamatergic cytotoxicity by stimulating microglia glutamate release and the inhibition of its transportation by astrocytes [34]. Glutamate is an excitatory neurotransmitter, and its accumulation leads to neuronal excitotoxicity [35]. Th overexposure to IL-1β, another major pro-inflammatory cytokine, can lead to substantial neuronal damage and worsen cognitive impairment by synaptic damaging [36,37]. The IL-1β derived from microglia cells causes synaptic alterations associated with cognitive impairment in sepsis, resulting in synaptic elimination and inhibition [38]. Moreover, high IL-1β levels are linked to the decrease of working and hippocampal-dependent memory [26]. Microglia production of NO, ROS, and derived reactive species molecules has deleterious effects on neurons, primarily inducing apoptosis or aggravating excitotoxicity, resulting in neurological deficits. The inducible nitric oxide synthase (iNOS) deficient septic mice have a cognitive impairment, suggesting its involvement in septic patients’ sequelae [39]. During sepsis, the deleterious amount of nitric oxide produced by the iNOS enzyme activity can cause microperforations in the BBB, decreasing CNS isolation by allowing the passage of plasma leakage [40]. Elevated levels of iNOS expression are also related with hypothalamic damage [41]. Thus, microglia activation has an essential role in sepsis. The exaggerated release of inflammatory mediators, NO production, oxidative stress, and glutamate secretion causes neuronal dysfunction, possibly leading to complications as SAE, long-term cognitive impairment, and sepsis-associated chronic pain [42,43] (Figure 1). 

The M2 phenotype, anti-inflammatory microglia, has a potential therapeutic role in diseases caused or aggravated by neuroinflammation [44,45]. Although neuroinflammation act to remove or inhibit brain-threatening pathogens, if sustained, it can induce neurotoxicity and neurodegeneration [46]. Neurodegenerative diseases are characterized by the chronic and progressive death of nerve cells in the brain and the spinal cord, altering the nervous system’s ability and functionality, with activated microglia further magnifying it with the release of cytotoxic factors [47]. In Alzheimer’s disease, M2 polarization improves the phagocytic ability of microglia, contributing to the clearance of amyloid-beta (Aβ), inhibiting its accumulation, reducing toxicity, and promoting neuroprotection [48,49]. An increase in the ratio of M2 microglia cells and the subsequent inhibition of the M1 profile is a prominent therapeutic target in multiple sclerosis (MS) and autoimmune encephalomyelitis (EAE), reducing the symptoms at the early stages of the disease, alleviating its progression [50]. The switch between microglia phenotypes is also a promising approach in treating Parkinson’s disease (PD), where it occurs progressive M1 overactivation, increasing the damage caused to the dopaminergic neurons [51]. M2 polarization is closely related to higher neuron survival rates in PD [52]. M2 microglia-derived BDNF plays an essential role in restoring the neuronal circuit after intracerebral hemorrhage (ICH), the most lethal stroke subtype [33,53]. The beneficial effects of targeting microglia changing phenotype can also be seen in the treatment of neuropsychiatric disorders, such as depression and PTSD, both affected by M1/M2 imbalance, with results showing behavioral improvement due to the inhibition of microglia-mediated neuroinflammation and higher presence of the M2 neuroprotective phenotype [54]. Different classes of transcription factors can tightly control microglia phenotypic diversity and function [55]. Table 1 highlights the neuroprotective role of M2 microglia. 

## 3. PPARγ-Dependent Anti-Inflammatory and Pro-Resolutive Mechanisms

Peroxisome proliferator-activated receptor γ (PPARγ) is a member of the nuclear receptor superfamily, being one of the three isotypes of PPARs, including also PPARα and PPARβ/δ. It usually forms a heterodimer with retinoid X receptor, either stimulated or repressed by recruited ligands, later binding to PPAR-responsive regulatory elements in the genome to control gene expression [56]. PPARγ plays a critical regulatory role in adipogenesis, lipid, and glucose metabolism [57], and it is a foremost regulator of the inflammatory response, shifting the immune system towards a more resolutive state, therefore decreasing the expression of pro-inflammatory cytokines [58]. Many immune cells express PPARγ, including monocytes/macrophages, neutrophils, dendritic cells, T and B lymphocytes, and platelets [59]. The activation of this nuclear receptor has a significant influence under macrophage phenotype, inducing the transition of M1 to M2 macrophages, and helping to keep the balance between them [60].

Similarly, PPARγ also regulates the microglia phenotype, leading to the polarization to the M2 cells [61]. The effects observed in macrophages and microglia phenotype and metabolism may occur due to the silencing of transcription factors, such as nuclear factor-κB (NF-κB), signal transducer, activator of transcription 1 (STAT-1), and activator protein-1 (AP-1) [62,63]. NF-κB plays a crucial role in stimulating inflammation, regulating innate and adaptive immune response aspects. Its activation contributes to the production of adhesion molecules, cytokines, and chemokines, and influences dendritic cell maturation, neutrophil survivor and recruitment, macrophage activation, and T cell activation and differentiation [64]. Given its wide range of functions and impact, an imbalance in the regulation of NF-κB can lead to devastating consequences, such as neurodegenerative disorders and chronic inflammation [65]. 

NF-κB is composed of five subunits (RelA/p65, RelB/p68, cRel/p75, p52, and p50) that bind to promoter regions of target genes as homodimers or heterodimers. These dimers remain in the cytoplasm in their inactive forms, connected to the inhibitory protein IκB family (IκBα, β, ε, γ, and δ) until further phosphorylation, disassociation, and subsequent nuclear translocation [66,67]. The subunits p65 and p50 form the most common heterodimer, with p65 playing a central role in inflammation activation by inducing various pro-inflammatory genes [68]. The NF-κB signaling pathways associated with MAPK activation induce the production of pro-inflammatory cytokines [69]. MAPK signaling pathways consist of a chain of proteins that transmit extracellular signals to intracellular targets through a series of activation and phosphorylation steps, essential to cellular regulation [70,71]. MAPK family (ERK, JNK, p38) is important for microglia inflammatory response [72]. The p38 MAPK signal pathways, for instance, exert a pivotal role in inflammatory and stress responses in various cell types, including microglia. Additionally, PPARγ inhibits p38 activation and p65 nuclear translocation, inhibiting NF-κB activity and preventing M1 polarization [73]. Thus, PPARγ potently suppresses NF-κB transcriptional activity, decreasing pro-inflammatory gene expression and the subsequent production of pro-inflammatory molecules, such as cytokines (TNF-α, IL-1, IL-2, IL-6, IL-8, and IL-12), chemokines (CXCL1 and CXCL10), and reactive nitrogen and oxygen intermediates [74,75]. Blocking ERK phosphorylation also led to the inhibition of NF-κB p65 phosphorylation and its nuclear translocation in microglia [76]. The overexpression of PPARγ inhibited the activation of both p38 and ERK, lowering M1 microglia activation, and increasing IL-10 expression, an indication of M2 presence [77]. Since NF-κB and the MAPK signal pathways are considered key effector contributors to M1 polarization, these data indicate the pivotal role that PPARγ has in M2 shifting. Additionally, PPARγ-deficient mice are incapable of M2 activation [78,79]. 

In sepsis, PPARγ has a pivotal role, with in vitro and in vivo showing beneficial effects in the inflammatory response control upon its activation [80]. Evidence shows that PPARγ activation decreases inflammatory and apoptotic levels, prolonging the survival rate in sepsis-induced acute lung injury [81]. PPARγ activation also attenuates liver dysfunction, one of the most vulnerable organs in sepsis, having a significant impact on the progression of the disease since liver metabolic functions are vital players in sepsis development [82]. Additionally, PPARγ diminishes sepsis-induced acute kidney injury, one of the most common complications observed during sepsis, in up to 70% of all patients [83]. PPARγ acute activation has a cardioprotective effect, downregulating the expression of pro-inflammatory cytokines, inhibiting apoptosis and necroptosis, ameliorating septic cardiac dysfunction [84], one of the leading causes of death in ICUs. This protective effect in sepsis is partially due to the PPARγ effects on macrophages, increasing their phagocytic capability, improving pathogen clearance, and mediating M2 polarization and resolution of inflammation [19,85,86,87]. 

Treatment with PPARγ agonists attenuates the inflammatory response, promoting a neuroprotective effect, reducing the injuries caused by neuroinflammation and oxidative stress in the brain [20,88,89]. The synthetic PPARγ agonists, known as a thiazolidinedione (TZD) or glitazones, are a class of antidiabetic drugs widely used experimentally as anti-inflammatory drugs due to their effects on cell proliferation, metabolism, and immune response [90,91].

Treatment with pioglitazone, a potent TZD PPARγ activator, reduced NF-κB activity and microglia M1-like behavior, protecting against dopaminergic neuron loss, thus attenuating motor dysfunction in a rat model of PD [92]. In combination with fenofibrate, a PPARα agonist, pioglitazone’s administration may exhibit a synergistic effect, contributing to ameliorating memory and cognitive impairment and reducing neuronal loss in a mouse model of Alzheimer’s disease [93]. Pioglitazone can also relieve depressive-like behavior in mice through PPARγ activation and the subsequent alteration in microglia phenotype, restoring the balance between pro- and anti-inflammatory cytokines [94]. A widely used PPARγ agonist rosiglitazone exhibits a protective effect over the blood–brain barrier (BBB) integrity by downregulating the expression of pro-inflammatory cytokines, such as TNF-α, IL-1β, and IL-6, mediators closely linked to the increase of BBB permeability [95]. The presence of the blood–neural barriers represents a challenge in drug delivery, significantly reducing their bioavailability in the CNS region [96]. The diffusion of lipid insoluble or larger than 4000 Da hydrophilic molecules is very limited through the BBB [97]. A few strategies minimize that problem using solid lipid nanoparticles (NPs), gold NPs, and polymeric NPs being the most prominent ones, successfully reducing adverse side effects, increasing drug concentration at the site of action, and consequently, improving the therapeutic response [97,98]. Nano-formulated rosiglitazone, for instance, has a neuroprotective effect lowering the levels of TNF-α and IL-6, upper regulating important neurogenesis growth factors, such as BNDF, glial cell line-derived neurotrophic factor (GDNF), and NGF, also improving memory and learning functions in a mice model of Alzheimer’s disease [98]. Table 1 exemplifies the PPARγ-dependent role of M2 microglia neuroprotective effects. 

PPARγ-induced neuroprotective effects result from both anti-oxidant and anti-inflammatory properties [99]. PPARγ plays a significant role as an anti-oxidant by means of a crosstalk with the transcription factor NF-E2 p45-related factor 2 (Nrf2), a well-known key down-regulator of the oxidative stress and inflammation as Nrf2 upregulates the expression of almost 200 cytoprotective genes. This gene expression leads to the production of many stress-responsive proteins, including glutathione, glutathione peroxidase, and superoxide dismutase (SOD), and protect the cells from oxidative and inflammatory stress. Nrf2 activating compounds modulates sepsis, being proposed as adjunct treatment as well [100,101,102].

PPARγ activation improves mitochondrial function in glial cells in neurological disease, activity which is relevant to SAE as well. TZDs improve mitochondrial oxidative phosphorylation and biogenesis in CNS. TZDs increased glucose utilization, lactate production, and mitochondrial membrane potential. Pioglitazone and rosiglitazone prevented the death of the neuroblastoma derived cell line SH-SY5Y cells because of mitochondrial biogenesis. These findings showed that PPARγ agonists are neuroprotective, increasing the neuronal survival by mitochondrial function improvement [103].

We must mention that TZD chronic use may cause side effects, such as bone loss, weight gain, and fluid retention, a concern in congestive heart failure, increased risk of myocardial infarction, and renal failure. More than 500 trials are in progress with the safest TZD in terms of side effects pioglitazone worldwide. Preliminary data showed pioglitazone is fairly secure for clinical use [90].

In addition to the synthetic compounds, natural PPARγ agonists represent a valid and promising option to regulate inflammation [104]. Natural products have made a real contribution to the history of pharmacotherapy, functioning as a reliable and diverse source of potential new drugs, especially in oncology and inflammation [105]. They act through numerous effective mechanisms, exhibiting anti-inflammatory and immunomodulatory properties, and nervous system protection and repairing [105]. Phytocannabinoid derivatives can function as PPARγ agonists, exerting anti-inflammatory and neuroprotective properties. Treatment with VCE-003.2, a cannabigerol derivative, reduced microglia reactivity in the substantia nigra, lowering the expression of pro-inflammatory markers, such as TNF-α, IL-1β, and iNOS, in a mice model of PD [106]. Curcumin, another natural compound that works as a PPARγ agonist, successfully promoted myelin formation, oligodendrocyte differentiation, and maturation, protecting these cells from inflammatory damage [107]. Curcumin-loaded nanoparticles possess higher bioavailability compared to free curcumin, and its administration reduced microglia and astrocyte activation and improved myelin repair, decreasing the extension of affected areas in a rat demyelination model [108]. Ursolic acid (UA), an herbal medicine with a wide range of pharmacological usage, has also been shown to act as a PPARγ agonist in the CNS. UA through PPARγ activation, induced the synthesis of the ciliary neurotrophic factor (CNTF) by astrocytes, which is involved in the differentiation and activation of these cells. UA also promoted neural repair by regulating oligodendrocyte progenitor cells differentiation, elevating remyelination, showing promising therapeutic potential in treating the chronic phase of MS [109]. Omega-9 is a PPARγ ligand, having a beneficial role in sepsis, helping to decrease pro-inflammatory cytokines and ROS production, and enhancing M2 macrophage polarization and anti-inflammatory cytokine secretion [110]. Our group showed that the pre-treatment with omega-9 prevented organ dysfunction and increased survival during experimental sepsis [111]. Furthermore, we showed that omega-9 improved bacterial clearance, possibly involving PPARγ, which contributed to a better sepsis outcome [112].

## 4. Critical View of the Actual Scenario

M1 and M2 differentiation are not as dichotomous as previously thought by the scientific community. Instead, microglia phenotype changes show a broader and continuous spectrum. The literature already presents evidence that these cells exert different functions in the CNS compared to macrophages differentiated from monocytes that infiltrate the region from the peripheral circulation. Studies suggest that those cells have distinct actions and exert microenvironment-dependent synergistic activity, increasing their spectrum of activities. Cytokines produced by microglia when activated for a pro-inflammatory profile are directly related to neuronal damage and the consequent cognitive sequelae observed in patients. The cognitive damage observed in septic survivors is a consequence of microglial overactivity. The secretion of cytokines and chemokines by glial cells causes an imbalance in neuronal homeostasis [80]. There is a strict correlation between the microglia M1 response and the process of neuroinflammation and neural damage, especially in neurodegenerative diseases. The currently available treatments still do not present satisfactory results, often only mitigating symptoms. The discussion about the neuroprotective role of the microglial M2 proves the importance of reestablishing the microglial physiological functions. There is an evident gap in developing new drugs and targets for patients with diseases that affect the CNS. Previous studies focused primarily on the neuron as the target for the action of new treatments [71]. Nowadays, microglia has been pointed out as having a protagonist role in CNS hemostasis control. Understanding the processes that lead to microglia’s activation and transition within a broad spectrum of phenotypic profiles will allow the discovery of new cellular and molecular targets and the proposal of more potent therapeutic strategies [10,20,21,23,25].

Microglia depletion experiments have revealed that there is microglial repopulation as a result. In the genetic microglial depletion model (*Cx3cr1^CreER^/Csf1r^flx/flx^*), microglia depletion occurs in a temporally controlled manner by tamoxifen injection intraperitoneally (acute model) or orally (chronic model). There was a clear microglial repopulation, showing that a microglial replacement with rapid microglial self-renewal occurred [113]. Microglial depletion in the nigrostriatal pathway of mice downregulated pro-inflammatory and anti-inflammatory gene expression in the PD model, showing microglial repopulation caused neuroprotection in PD mice [114]. Th microglia that repopulates can restore microglial homeostatic functions after microglial depletion, which is a new proposal of therapeutic strategy for AD as well [115,116].

The effects of sepsis on the nervous system are well known and currently represent a challenge for physicians. Although exacerbated dysregulated inflammatory response causes neuronal damage due to the neuroinflammation process, a few studies focus on microglia as a possible therapeutic target for sepsis resolution and better outcomes [38,39]. Sepsis causes increased susceptibility to Aβ oligomers, as it occurs in AD. Microglia from septic mice shift morphology to amoeboid/phagocytic when exposed to low amounts of AβO, and increase of pro-inflammatory proteins. Microglial depletion with either minocycline or colony stimulating factor 1 receptor inhibitor (PLX3397), impaired cognitive dysfunction induced by AβO during experimental sepsis [117]. However, the microglial phenotype after those depletion experiments remains unknown. We have highlighted the importance of microglia and its potential for developing new treatments, thus promoting new studies that increasingly elucidate their role in the pathophysiology of sepsis. The neuroprotective role of M2 microglia in studies involving neurodegenerative diseases and pathophysiological changes that generate brain damage as a consequence can be implied in sepsis in further studies.

Microglia crosstalk with astrocytes contributes to CNS homeostasis. Astrocytes act in the brain during sepsis, controlling with the microglia, the severity of SAE and cognitive impairment. Modulating astrocyte activity and the crosstalk with microglia towards M2 may help in neuroinflammation management and block brain dysfunction [118]. The cognitive impairment and brain dysfunction associated with SAE directly correlate with higher mortality rates, and most of the reports indicate the crucial role that neuroinflammation characterized by the inflammatory cytokine storm plays in pathophysiology. Microglial activation is an important cellular target of SAE, and inhibition of microglia response has improved long-term cognitive behavior, reducing exaggerated neuroinflammation in mice undergoing experimental sepsis [79,84]. As there are no specific treatments for SAE, modulation of microglial phenotypes may be a prominent therapeutic strategy.

Our review suggests PPARγ as a molecular target for studies involving the effects of sepsis in the SNC, based on its anti-oxidant, anti-inflammatory and pro-resolutive properties, which are already well established in the literature for many diseases. PPARγ agents are better than the conventional treatment for SAE, that includes fluid, and non-pharmacological approaches [39]. The broad spectrum of PPARγ agent activities reestablishing homeostasis in various mechanisms show its superiority over the commonly clinically used therapy of SAE. So far, there is no specific treatment for SAE [39].

We pointed out the PPARγ role on microglial polarization, thus highlighting its neuroprotective properties. Th importance of the PPARγ activation and its therapeutic potential in several neurological and neurodevelopmental disorders has already been demonstrated, as PPARγ’s role in mitigating the neuroinflammatory process. PPARγ inhibition of the oxidative stress, inflammatory response, and prompting of resolutive processes reduces the M1 microglia and increases the M2 microglia in the CNS. PPARγ activation restores the balance between microglial phenotypes, which is essential for the reestablishment of homeostasis and the reduction of symptoms observed in these patients. Once again, although there are already studies on PPARγ activation in sepsis, including our own articles [119,120], little is known about PPARγ in microglia during infection, especially when trying to mitigate SAE. Furthermore, PPARγ activation may reduce neuronal death, thus improving the life quality of patients who recovered from sepsis, who often live with sequelae for long periods.

The activation of PPARγ and the consequent inactivation of intracellular signaling proteins and transcription factors, such as MAPK and NF-kB, can attenuate inflammatory processes. As we recall, both MAPK (ERK and p38) and NF-kB are essential for initiating and maintaining the inflammatory process, playing a fundamental role in the differentiation of microglia into the M1 phenotype. The MAPK signaling pathway is involved in the inflammatory process through the activation of transcription factors and the expression of pro-inflammatory genes, resulting in increased production of cytokines, such as TNFα and IL-6. Conversely, the inhibition of ERK or p38 proteins, and the transcriptional activity of NF-kB results in a reduced presence of M1 microglia, thus reducing the impact of its activity on nervous tissue [34,73].

Elucidation of molecular mechanisms by which PPARγ acts in the phenotypic change of microglia, while highlighting its importance in the control and resolution of the inflammatory process, thus increases the demand for studies that analyze the fundamental mechanisms that these cells use in response to the infectious agent. In addition to fundamental knowledge of interplay among intracellular players, it is urgent to develop new therapeutic adjuvant strategies as new drugs that affect those molecular pathways or even drug repositioning, expanding the possibilities for EAS treatment.

## 5. Final Remarks

The cognitive impairment and diffuse cerebral dysfunction associated with SAE directly correlate with higher mortality rates, with most of the reports indicating the crucial role that oxidative stress, neuroinflammation and inflammatory cytokine release have in the pathophysiology. Microglia activation is an essential cellular component of SAE, and its inhibition improved long-term cognitive behavior, reducing exaggerated neuroinflammation in CLP mice. There are no specific treatments for SAE; therefore, the modulation of microglial phenotypes could be a prominent therapeutic strategy. PPARγ anti-oxidant, anti-inflammatory, and pro-resolutive properties are well established. PPARγ agonists may be considered potential candidates for drug repurposing, potent drugs for the adjuvant treatment of SAE in association with antibiotics. PPARγ effects on microglia polarization highlights its neuroprotective properties and PPARγ therapeutic potential in treating several neurological and neurodevelopmental disorders, thus, showcasing the PPARγ role in mitigating degenerative sterile and infectious disease processes in the brain and peripheral systems.

## Figures and Tables

**Figure 1 ijms-23-04925-f001:**
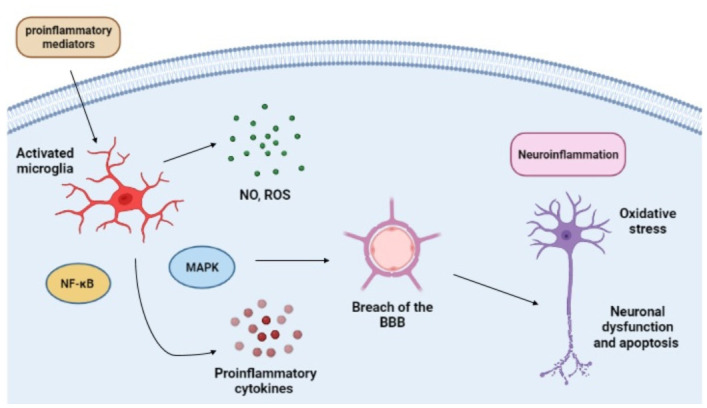
Simplified scheme of microglia activation and its role in the damaging of neurons during sepsis. Created with BioRender.com.

**Table 1 ijms-23-04925-t001:** The neuroprotective role of M2 microglia.

Study	Model	Main Results
Peng, Jing et al., 2019 [19]	Status epileptic male C57BL/6 mice aged 8–10 weeks	Polarization of microglia to M2 by PPARγ ligand rosiglitazone and protection against pilocarpine-induced status epilepticus with rescued neuron loss
Wen, Liang et al., 2018 [20]	Traumatic brain injury in C57BL/6J mice (10 to 12 weeks old)	M2 microglia attenuated axonal injury in the cerebral cortex and improved neurological function
Qin, Xiaqing et al., 2020 [45]	Chronic unpredictable mild stress in male wt mice (23–25 g, 8–10 weeks old) and male ob/ob mice (43–53 g, 8–10 weeks old)	The behavioral improvement due to the inhibition of microglia-mediated neuroinflammation and higher presence of the M2 neuroprotective phenotype in mice treated with the PPARγ agonist pioglitazone
Ren, Chaoxiu et al., 2020 [47]	Alzheimer disease in female APP/PS1 double transgenic mice (6 months old)	M2 microglia degraded Aβ deposits and efficiently promoted neuroprotection by inhibiting Aβ accumulation and neuroinflammation
Xie, Zhishen et al., 2020 [48]	Alzheimer-like disease in transgenic *C. elegans*	M2 microglia enhanced Aβ degradation reducing its deposition in the PPARγ-dependent mechanism
Zhang, Youwen et al., 2018 [50]	MPTP-intoxicated male C57BL/6 mice (5–6 weeks, weight 18–22 g) Parkinson disease model	Higher levels of M2 microglia alleviated neuroinflammation
Bok, Eugene et al., 2018 [51]	LPS-lesioned inflammatory model of Parkinson disease in female Sprague Dawley rats (230–280 g)	M2 microglia enhanced the survival of dopamine neurons
Miao, Hongsheng et al., 2018 [52]	Intracerebral hemorrhage in Sprague Dawley (SD) rats (250–350 g)	M2 microglia-derived BDNF promoted neurogenesis

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
