# Peer review of "Polarization of Microglia and Its Therapeutic Potential in Sepsis"

_ijms, 2022, doi:10.3390/ijms23094925_

Round 1

Reviewer 1 Report

the presented document is well developed and described

the figure and table are correct

on inflammatory processes, they are always linked to oxidative stress. a regulator of both processes is Nrf2, can you include this point?

Thanks

Author Response

Reviewer 1

the presented document is well developed and described

the figure and table are correct

On inflammatory processes, they are always linked to oxidative stress. a regulator of both processes is Nrf2, can you include this point?

Answer: Thanks for the nice words. We included the missing information about the association between oxidative stress and inflammation and the essential role of Nrf2 as a regulator of both processes (page 10).

Reviewer 2 Report

This is an interesting review paper that discusses the utility of utilising  PPARγ in neuroprotection in Sepsis and its role in microglia polarization and anti-inflammatory properties.

The paper is well written and appropriately detailed. I feel that the discussion of other glial cells such as astrocytes would be appropriate in sepsis as well as more information on nitric oxide synthase.

The use of PPAR agonists is appropriate but in view of the microglial focus of the paper some discussion about compounds that can cross the blood brain barrier may be appropriate as well as a their pleotropic action and potential ability to up-regulate the antioxidant status and mitochondrial function of neurons and glial cells which is relevant to sepsis given the involvement of these parameters in this disorder.

It should be stressed that these PPAR agonist treatments are only only adjunct therapies and potential side effects should be highlighted.

Author Response

Reviewer 2

(x) Moderate English changes required

Answer: We have carefully revised the manuscript for grammatical and style English errors.

This is an interesting review paper that discusses the utility of utilising PPARγ in neuroprotection in Sepsis and its role in microglia polarization and anti-inflammatory properties.

The paper is well written and appropriately detailed. I feel that the discussion of other glial cells such as astrocytes would be appropriate in sepsis as well as more information on nitric oxide synthase.

Answer: Thanks for the nice words about our review. We discussed the role of astrocytes (page 12) in sepsis as well as added more information on nitric oxide synthase (page 4).

The use of PPAR agonists is appropriate but in view of the microglial focus of the paper some discussion about compounds that can cross the blood brain barrier may be appropriate as well as a their pleotropic action and potential ability to up-regulate the antioxidant status and mitochondrial function of neurons and glial cells which is relevant to sepsis given the involvement of these parameters in this disorder.

Answer: We discussed compounds that can cross the blood brain barrier (pages 9 and 10), the antioxidant status and mitochondrial function (page 10). We believe those references added important information about the pleiotropic action of PPAR gamma agonists.

It should be stressed that these PPAR agonist treatments are only only adjunct therapies and potential side effects should be highlighted.

Answer: We stressed that PPAR agonist treatments are only adjunct therapies and highlighted potential side effects (page 10).

Reviewer 3 Report

Sepsis-associated encephalopathy (SAE) is  the most frequent sepsis-associated organ dysfunction, with symptoms such as deliriums, seizures,  and comma, linked to increased mortality, morbidity, and cognitive disability. However, the current therapy does not avoid those patients' symptoms. The authors presented a microglial polarization as a therapeutic target of SAE and they highlighted the `PPAS g as a molecular target for the treatment of SAE. I think the manuscript includes new and intriguing findings, however the authors should revise it according to the following concerns;

  1. The authors should describe why the authors focused on the `PPAS g as a molecular target for the treatment of SAE.
  2. The authors should describe the superiority of PPAS g agents over the conventional therapy for the treatment of SAE, citing relevant literatures.

Author Response

Reviewer 3

(x) English language and style are fine/minor spell check required

Answer: We have carefully revised the manuscript for minor syntaxis/grammatical English errors.

Sepsis-associated encephalopathy (SAE) is the most frequent sepsis-associated organ dysfunction, with symptoms such as deliriums, seizures, and comma, linked to increased mortality, morbidity, and cognitive disability. However, the current therapy does not avoid those patients' symptoms. The authors presented a microglial polarization as a therapeutic target of SAE and they highlighted the `PPAS g as a molecular target for the treatment of SAE. I think the manuscript includes new and intriguing findings, however the authors should revise it according to the following concerns;

  1. The authors should describe why the authors focused on the `PPAS g as a molecular target for the treatment of SAE.
  2. The authors should describe the superiority of PPAS g agents over the conventional therapy for the treatment of SAE, citing relevant literatures.

Answer: We improved the arguments favoring the focus on the PPAR g as a molecular target for the treatment of SAE along with the text and reinforced that PPAR g agents are a better choice of adjunct therapy when compared to the conventional treatment for the SAE treatment, adding more references. We believe we have strengthened our main point. We appreciate the comments that helped us to improve the manuscript.

This manuscript is a resubmission of an earlier submission. The following is a list of the peer review reports and author responses from that submission.